# Osteogenesis of 3D-Printed PCL/TCP/bdECM Scaffold Using Adipose-Derived Stem Cells Aggregates; An Experimental Study in the Canine Mandible

**DOI:** 10.3390/ijms22115409

**Published:** 2021-05-21

**Authors:** Joon Seok Lee, Tae Hyun Park, Jeong Yeop Ryu, Dong Kyu Kim, Eun Jung Oh, Hyun Mi Kim, Jin-Hyung Shim, Won-Soo Yun, Jung Bo Huh, Sung Hwan Moon, Seong Soo Kang, Ho Yun Chung

**Affiliations:** 1Department of Plastic and Reconstructive Surgery, School of Medicine, Kyungpook National University, Daegu 41944, Korea; leejspo@knu.ac.kr (J.S.L.); taehyeon1021@gmail.com (T.H.P.); rjyflying@naver.com (J.Y.R.); fullrest74@hanmail.net (E.J.O.); sarang7939@naver.com (H.M.K.); 2TINA Aesthetic Surgical Clinic, Daegu 41938, Korea; conyong@naver.com; 3Cell & Matrix Research Institute, School of Medicine, Kyungpook National University, Daegu 41944, Korea; 4Department of Mechanical Engineering, Korea Polytechnic University, 237 Sangidaehak-Ro, Siheung-si 15073, Gyeonggi-do, Korea; happyshim@kpu.ac.kr (J.-H.S.); wsyun@kpu.ac.kr (W.-S.Y.); 5Research Institute, T&R Biofab Co., Ltd. 242 Pangyo-ro, Seongnam-si 13487, Gyeonggi-do, Korea; neoplasia96@hanmail.net; 6Department of Prosthodontics, Dental Research Institute, Institute of Translational Dental Science, School of Dentistry, Pusan National University, Yangsan-si 50612, Korea; sunghwanmoon@tnrbiofab.com; 7College of Veterinary Medicine, Chonnam National University, Gwangju 61186, Korea; vetkang@chonnam.ac.kr; 8BK21 FOUR KNU Convergence Educational Program of Biomedical Science for Creative Future Talents, School of Medicine, Kyungpook National University, Daegu 41944, Korea

**Keywords:** adipose-derived stem cells, aggregate, osteogenesis, 3D-printed PCL/TCP/bdECM scaffold

## Abstract

Three-dimensional (3D) printing is perceived as an innovative tool for change in tissue engineering and regenerative medicine based on research outcomes on the development of artificial organs and tissues. With advances in such technology, research is underway into 3D-printed artificial scaffolds for tissue recovery and regeneration. In this study, we fabricated artificial scaffolds by coating bone demineralized and decellularized extracellular matrix (bdECM) onto existing 3D-printed polycaprolactone/tricalcium phosphate (PCL/TCP) to enhance osteoconductivity and osteoinductivity. After injecting adipose-derived stem cells (ADSCs) in an aggregate form found to be effective in previous studies, we examined the effects of the scaffold on ossification during mandibular reconstruction in beagle dogs. Ten beagles were divided into two groups: group A (PCL/TCP/bdECM + ADSC injection; *n* = 5) and group B (PCL/TCP/bdECM; n = 5). The results were analyzed four and eight weeks after intervention. Computed tomography (CT) findings showed that group A had more diffuse osteoblast tissue than group B. Evidence of infection or immune rejection was not detected following histological examination. Goldner trichrome (G/T) staining revealed rich ossification in scaffold pores. *ColI*, *Osteocalcin*, and *Runx2* gene expressions were determined using real-time polymerase chain reaction. Group A showed greater expression of these genes. Through Western blotting, group A showed a greater expression of genes that encode ColI, Osteocalcin, and Runx2 proteins. In conclusion, intervention group A, in which the beagles received the additional ADSC injection together with the 3D-printed PCL/TCP coated with bdECM, showed improved mandibular ossification in and around the pores of the scaffold.

## 1. Introduction

3D printing technology has contributed to significant advancements in many different fields over the past 20 years. Many countries have attempted to develop the 3D printing industry for business, fashion, and mechanical engineering [1,2,3]. With the recent advances in 3D printing technology and increasing research in biomedical fields, 3D printing technology is seen as a new tool of change, especially in biotechnology involving tissue engineering and regenerative medicine. In fact, developing artificial organs and tissues using 3D printing technology has been recently reported [4].

The tissue engineering area has seen active research on manufacturing artificial scaffolds using various materials. It aims to restore, maintain, and improve the damaged function of the living body. To perform such recovery and regeneration of tissues, materials required for the manufacturing of scaffolds must have biocompatibility, biodegradability, and mechanical properties that can be maintained in the body. Additionally, cells injected into and attached to the pores manufactured according to the characteristics of the implantation site of the scaffold must be able to differentiate and proliferate. These internal connective structures through the pores show excellent biocompatibility by enhancing the penetration, differentiation, and proliferation of cells [5,6,7,8].

Manufacturing of scaffold for tissue engineering through the 3D printing technology allows patient-customized scaffolds based on patients’ computed tomography (CT)/magnetic resonance imaging (MRI) data. These scaffolds have excellent internal connectivity because of the 3D shape. Their mechanical properties, pore size, and porosity can be controlled [9,10,11,12,13,14].

The extracellular matrix (ECM) provides important clues for long-term cell proliferation and differentiation and creates a microenvironment consisting of cell–cell connectivity and 3D cells [15]. ECM is a sophisticated biomaterial composed of various collagens, non-collagen proteins (NCPs), and proteoglycans. These secreted collagens combine with NCP to form a dense structural hierarchy. Then, amorphous and non-crystalline ECM is converted into a more crystalline form and undergoes bone mineralization for increased hardness and strength of the tissue [16]. Previous studies have demonstrated the important role of NCP in bone mineralization.

Recent studies have shown that demineralized bone matrix (DBM) causes more inflammation than synthetic hydroxyapatite compounds due to using additional viscous carriers, such as water-soluble polymers (sodium hyaluronate or carboxymethylcellulose) or anhydrous aqueous solvents (glycerol). Therefore, bone-demineralized and decellularized extracellular matrix (bdECM) has been developed to make a gel form of the same biological material without using an additional solvent [17]. This bdECm hydrogel has the same bone conductivity and fluidity [18]. Therefore, bdECM has been manufactured using ECM extracted from animal bones and has been widely used in bone tissue engineering. A 3D printing technique in which bdECm is added to a polycaprolactone/tricalcium phosphate (PCL/TCP) porous scaffold has recently been developed to promote bone regeneration and improve adhesion and proliferation of osteoblasts. Interestingly, using bdECm as a bio-ink for bioprinting of cells has been raised recently [19,20,21].

Stem cells refer to those cells capable of self-renewal and differentiation into various types of cells under appropriate biological signals and external stimuli. Mesenchymal stem cells (MSCs) from fetal tissues, umbilical cord blood, and bone marrow and adipose-derived stem cells (ADSCs) can be differentiated into various tissues, such as bone, cartilage, muscle, and nerve through tissue engineering [22,23,24,25,26]. Several studies have reported bone differentiation after the injection of stem cells into the 3D scaffold. However, most studies reported limitations in bone differentiation and adhesion of stem cells to the scaffold. Many studies are currently being conducted to increase adhesion and bone differentiation by using aggregates of ADSCs [27,28].

This study was conducted to assess the effects of 3D-printed polycaprolactone (PCL)/tricalcium phosphate (TCP)/bdECM biomaterials and ADSC aggregates on the ossification of mandibular bone reconstruction in beagles. bdECM was coated on PCL/TCP scaffold, which was used to promote ossification in a previous study. We hypothesized that ADSC aggregates would increase bone formation inside the contact surface and scaffold in forming the bone tissue [29,30,31,32].

## 2. Results

### 2.1. Outcomes of the Evaluation Using CT

#### CT Results: Coronal, Axial, and Sagittal Views

In both groups, ossification was not observed at four weeks on CT compared to normal bones. At week 8, the density was increased on CT images, similar to that of normal bones alongside the margins of the scaffold, unlike the previous pattern. This was the pattern of ossification progression alongside the marginal region. The density of CT images increased more markedly at eight weeks than at four weeks. As a result, the findings suggested that ossification of the marginal region continued to increase at eight weeks.

In both groups, the size and number of internal pores were decreased at eight weeks. However, the bone density of the scaffold was not as high as the surrounding normal bone tissues and only showed a partial increase. This suggests that most of the pores were products of inflammatory reactions with surrounding tissues and an increased number of connective tissues. Only partial ossification occurred. At eight weeks, the pores were further decreased, and bone density was also higher than that at four weeks.

The PCL/TCP/bdECM + ADSC aggregate administration (PTE + SA) group showed a more pronounced marginal ossification and a greater decrease in pores compared to the PCL/TCP/bdECM (PTE) group (Figure 1).

Bone density was measured using the Hounsfield unit (HU) in five grafts of each group. The average HU was compared between the two groups to measure the density in the pores. In the PTE + SA group, the density was 337.28 at week four and 372.32 HU at eight. In the PTE group, the density in the pores was 248.12 HU at four weeks and 273.38 HU at eight weeks.

Although there was a difference in HU between the two groups, HU increased over time, suggesting that ossification progressed sufficiently. Additionally, the PTE + SA group showed a higher density than the PTE group, confirming that ossification was progressing better (Figure 2).

### 2.2. Histological Findings

Biopsies were obtained from each group in four locations to observe changes in inflammation and connective tissues, including collagen fibers, as well as the extent of ossification through G/T stain. In both groups, periosteal connective tissue was enclosed in the marginal region of the scaffold, and a dense pattern of connective tissues was also observed inside the pores. These suggested that the scaffold had high biocompatibility. Inflammation was partially observed in the pore, which seems to be caused by inflammatory reactions before engraftment of the scaffold. Ossification was observed in each marginal area, and the PTE + SA group had higher ossification than the PTE group. Biopsy of the center of the scaffold showed a relatively more distinct ossification pattern than that of the marginal region. These findings suggested that ossification was better induced in the PTE + SA group administered with ADSCs than in the PTE group (Figure 3 and Figure 4).

### 2.3. RT–PCR

At eight weeks after the 3D-printed model surgery, ossification was observed in the scaffold pores with increased fibrinogen around the tissue. The increase in fibrinogen was confirmed by the increased expression of collagen type I (*ColI*) gene, and ossification was confirmed by the increased expressions of *Osteocalcin* and *Runx2* genes. Real-time polymerase chain reaction (RT–PCR) results showed that the levels were slightly increased in the PTE + SA group, which was administered ADSCs, compared to the PTE group (Figure 5).

### 2.4. Western Blot

At eight weeks after 3D-printed model surgery, the expression of proteins related to ossification was increased in both groups. Runx2, Col, and Osteocalcin expressions were also increased. The band intensity of COL1, OCN, and RUNX2 was normalized to that of β-actin. Western blot results indicated that the protein levels were increased in the PTE + SA group administered with ADSCs compared to those in the PTE group (Figure 6).

## 3. Discussion

3D printing technology has rapidly developed through many studies and is now used in various aspects of medicine due to its convenience and suitability. Recent studies also show that 3D printing technology can be used in studies on bones, ears, exoskeletons, respiratory organs, jawbones, lenses, cell culture, stem cells, blood vessels, vascular networks, tissues, and organs [33,34,35]. However, the lack of specialized software for simulation, an insufficient correlation between preoperative simulation and actual surgery, problems of accuracy, and the possibility of artifacts in data obtained through CT scans limit the further use of 3D-printed models. Despite these limitations, 3D printing technology is still recognized as a new medical technology, and soon, it is expected to help medical staff to visualize and specify various characteristics of each patient [36,37,38].

On the other hand, tissue engineering is also opening a new era in bioengineering. In general, tissue engineering includes three components: cells, scaffolds, and growth factors. 3D technologies have been used recently in cell and tissue printing techniques. Although there are numerous challenges to overcome, 3D technology is expected to surpass the existing traditional cell culture technologies in tissue engineering. It is expected that it would be possible to simultaneously form living cells and scaffolds using 3D living cell printing technology [1,39,40,41,42].

Scaffolds in tissue engineering are generally made of natural and synthetic polymer materials. Natural polymer materials extracted from natural substances, animals, and humans have superior biocompatibility compared to other materials and are nontoxic. These include gelatin, collagen, fibrin, elastin, and alginate. Synthetic polymer materials are relatively inexpensive, have outstanding mechanical properties, and are hydrolyzed in vivo or decomposed by enzymes. Therefore, these are ideal polymers as a supporter. Examples of synthetic polymer materials include poly-ε-caprolactone (PCL), poly-lactide-co-glycolide (PLGA), and polylactic acid (PLA). In particular, various scaffolds for bone tissue regeneration are manufactured using polylactide, a polymer material used for bone tissue regeneration, and TCP, a bioceramic material. Most recently, resorbable membranes have been utilized preferentially because the non-resorbable forms inevitably require a surgical procedure for membrane removal, which can cause further patient discomfort, risk of tissue damage, and additional costs and duration of treatments. Accordingly, studies on synthetic bioresorbable materials have been conducted for the fabrication of form-stable resorbable GBR membranes with a sufficient degradation rate. Bioresorbable materials, such as polycaprolactone (PCL), polyglycolides (PGAs), polylactides (PLAs), and copolymers, have been used for medical purposes [43].

A previous study fabricated thin-membrane-type scaffolds blending polycaprolactone (PCL), poly(lactic-co-glycolic acid) (PLGA), and beta-tricalcium phosphate (β-TCP). It confirmed that the PCL/PLGA/β-TCP membrane prepared using the 3D printing technology promoted appropriate bone-formation in a rabbit calvaria bone-defect model. This bioresorbable PCL/PLGA/β-TCP membrane has the biological and mechanical advantages of both PCL and PLGA, as well as the osteoconductivity of TCP. However, PLGA also has been reported to induce an inflammatory response because of the acidic byproduct and toxins produced during its degradation process. Therefore, this study was undertaken by only using PCL/TCP [44].

However, unlike polymer materials that are solid and produced using heat, bioceramic materials are in powder form. They have lower mechanical strength than scaffolds made of polymers. In addition, the material is in powder form, which leads to difficulties in manufacturing a 3D support with excellent internal pores. Therefore, many studies are being conducted to mix ceramic and polymer materials.

In general, scaffolds must be strong enough to resist external forces. They must be disassembled after the pores are filled with bones at an appropriate time. In our study, a scaffold was manufactured using PCL/β-TCP, a popular biodegradable material with osteoinductive properties used in many different clinical studies. β-TCP has a high affinity for BMP-2, which is a factor required for bone production. Thus, it induces bone formation and β-TCP biodegrades over time. These two important features are ideal for a scaffold [45,46,47,48,49].

ECM contains collagen, non-collagen proteins, and proteoglycans, all of which play an important role in cell proliferation and differentiation by providing a microenvironment through adequate intercellular connections. In the recent study [15], it was confirmed that extracellular matrix (ECM) composes a favorable environment for long-term cell proliferation and differentiation in the scaffold. Amorphous and non-crystalline ECM is converted into a more crystalline form and undergoes bone mineralization for increased hardness and strength of the tissue [16]. However, it is challenging to develop biomaterials by imitating the ECM composition of the target tissue [19,50]. For this reason, DBM produced using ECM extracted from bovine bone has been widely used for the engineering of bone tissues [51,52].

DBM contains growth factors, collagen and non-collagen proteins. Thus, it reproduces the microscopic habitat environment, which enables osteoconduction and osteoinduction [17,53]. However, DBM xenografts may trigger immune responses. Therefore, bdECM gels manufactured without using additional solvents, while having the same bone conduction and induction properties, have been used [18,20].

In recent studies, a 3D printing technique that adds bdECM to a PCL/PLCA/β-TCP porous scaffold was developed to promote bone regeneration and improve adhesion and proliferation of osteoblasts. There are also reports that bdECM can be used as a bio-ink necessary for the bioprinting of cells [19,20]. Based on these studies, Bae et al. observed that bdECM in 3D-printed scaffold increased initial cell adhesion and improved ALP. In other words, a scaffold with bdECM containing rhBMP-2 promotes skeletal differentiation [54,55,56,57]. Accordingly, in this study, bone-demineralized and decellularized extracellular matrix (bdECM) using ECM was additionally used in the scaffold. Through this, the adhesion and proliferation of the bone regeneration and osteoblast increased. Due to this nature, bdECM is also widely applied as bio-ink the bioprinting process.

Stem cells capable of differentiating from adipose tissue into other tissues were discovered in 2001 by Zuk and Huang et al. The isolation process of ADCS is relatively simple compared to that of bone marrow-derived MSCs (BM-MSCs). Adipose tissue is abundant throughout the body, and a greater number of cells can be isolated. Additionally, there are no ethical problems, unlike embryonic stem cells. Isolation of BM-MSCs can lead to possible complications, such as pain and infection, and the amount of BM that can be collected is limited. On the other hand, ADSCs can be collected in large quantities under local anesthesia. Therefore, they have been widely used to promote bone fusion through tissue engineering in bone grafts and the regeneration of bone defects [22,23,24,25].

The use of ADSCs aggregates, which affects bone formation, for 3D-printed scaffold has several advantages. Scaffolds are good mediators for the survival and bone formation of these ADSCs [58,59].

A difference from other studies is the method of injecting adipose-derived stem cells (ADSCs) within the scaffold. ADSCs can be differentiated into various tissues, such as the bone, cartilage, muscle, and nerves, through tissue engineering. Several studies have reported bone differentiation after the injection of stem cells into the 3D scaffold. In the earlier study, cultured cells were seeding or injected in the scaffold as it is.

However, in this study, we gathered ADSC aggregate that was created and injected into the scaffold. This can heighten the cell density much more significantly, so the paracrine effect can be enhanced. Through the paracrine factors, we confirmed a result with increased cell proliferation and bone regeneration.

To evaluate the bone regeneration ability of ADSC aggregates and bdECM-coated PCL/β-TCP scaffold, micro-CT and histological staining were analyzed after implanting the scaffold into the mandibular defects of beagle dogs. It was observed that the volume of new bone increased in both groups. More new bone was observed in the PTE + SA group than in the PTE group. RT–PCR and Western blot showed increased expression of bone proteins in both groups, and the increase was higher in the PTE + SA group than in the PTE group. These findings suggested that 3D-printed PCL/β-TCP/bdECM biomaterial and ADSC aggregates were more effective in reconstructing the mandibular defect of beagle dogs. Unlike in previous studies, macroscopical and histological results demonstrated better outcomes in those injected with ADSC aggregates after coating of bdECM.

Despite various contents and methods, the follow-up period after surgery was short (eight weeks). As a result, possible chronic immune rejections could not be evaluated. Second, the scaffolds were undissolved until eight weeks after surgery. Ossification in the bone healing process can replace the biomechanical properties of the scaffold. Assessment of the optical density through tissue slides showed that the optical density of the tissue inside the pores was significantly increased. However, the relative optical density in the pores was always less than that of the normal bone tissue. Compared to the surrounding periosteum, the optical density was greater and lesser in different cases. Additionally, relatively different results were observed at the same site depending on the differences between G/T and hematoxylin and eosin (H/E) staining. A certain level of variation may have occurred as the measurement site was randomly chosen in an arbitrary size.

## 4. Materials and Methods

### 4.1. Materials

#### 4.1.1. 3D Printing Scaffold Manufacturing of PCL/TCP/bdECM

Biodegradable PCL (Evonik Industries, Germany) was dried sufficiently at 105 °C for one day to make a hydrophilic polymer. TCP power (Sigma-Aldrich Co., St. Louis, MI, USA) was manufactured with a particle size of 100 nm or less.

A 3D-printed scaffold of PCL/TCP was manufactured using a heating jacket and stainless-steel cylinders through micro-nozzles along the X–Y–Z-axis. IMS computer software (Pohang University of Science and Technology, Korea) was used as the production system. Fused Deposition System was used for designing the 3D-printed scaffold (Figure 7).

For coating of bdECM, porcine bone was freeze-dried at −85 °C for 24 h to adjust the size particles of porcine bone (SPB). Then, 70% ethanol was used to wash off fat, and it was dechlorinated with 0.5 N hydrochloric acid (HCL). The HCL solution was replaced every two hours, three times in total, to remove impurities. The decontaminated SPB was washed three times with distilled water (DW), put in a solution containing 0.05% trypsin and 0.02% EDTA and kept at 37 °C for 2 h, and washed again with DW. This process was repeated three times. Then, SPB was freeze-dried at −85 °C for 24 h and powdered in a freeze-mill (6875D, SPEX SamplePrep, Metuchen, NJ, USA). The porcine bone was then dissolved in an acidic pepsin solution to be manufactured into a bdECM gel coated on a 3D-printed scaffold of PCL/TCP.

#### 4.1.2. Identification of ADSCs and Aggregates Formation

The adipose tissue collected from the abdominal cavity of a beagle dog was washed with the same volume of phosphate-buffered saline (PBS) before tissue degradation. The adipose tissue was then degraded using 0.075% collagenase type I (Worthington Biochemical, Lakewood, NJ, USA) for 30 min at 37 °C. The decomposed tissue was filtered to remove debris from the connective tissue. After separation of the fat cell layer floating on the upper layer, the cell suspension was centrifuged for 10 min at 200 G. Contaminated red blood cells were removed by adding erythrocyte lysis buffer of pH 7.3. The stromal cells were washed twice with PBS, and ADSCs were collected. The ADSCs were cultured in Dulbecco’s modified Eagle’s medium (Thermo Fisher Scientific, Waltham, MA, USA) supplemented with 10% fetal bovine serum (FBS) (Hyclone, Logan, UT, USA) and 1% penicillin and streptomycin in a CO_2_ incubator at 37 °C. ADSCs cultured were incubated at 37 °C exposed to 5% CO_2_ in an incubator for 24 h and 4 weeks, respectively. The ADSCs were treated with a trypsin-EDTA enzyme to separate them into single cells. They were then suspended in 0.2 mL microtubes (8-strip PCR tube) and centrifuged at 3000 rpm for five minutes to collect cell aggregates.

#### 4.1.3. Experimental Animals and Groups

Ten 36-months-old healthy beagles were selected. Before the 3D-printed model experiment, spiramycin and metronidazole were injected into the blood vessels under general anesthesia. Tooth scaling was performed and left molar and premolar teeth were removed.

The beagles were assigned into two groups (N = 5 for each group) based on the application method of the adipose stem cell aggregate before implantation of the 3D-printed PCL/TCP/bdECm biomaterial.

PCL/TCP/bdECM + ADSC aggregate administration group (PTE + SA group): 3D-printed PCL/TCP/bdECM biomaterial was placed and fixed in the mandibular defect of beagles. Then, 0.1 mL of the adipose stem cell aggregate at a concentration of 1 μg/mL was added to the 3D-printed PCL/TCP/bdECM biomaterial through 10 pores distributed across two rows at regular intervals using a micropipette. 0.01 mL of ADSC aggregates was administered to each pore.

PCL/TCP/bdECM group (PTE group): Unlike the PTE + SA group, 3D-printed PCL/TCP/bdECM biomaterial was placed and fixed in the mandibular defect of beagles, and ADSC aggregates were not administered.

### 4.2. Experimental Methods

Thiopental sodium (Pentothal, Choongwae Parma Co., Seoul, Korea) was intravenously injected for anesthesia during surgery, followed by general anesthesia using Halothane (Ilsunghalothane, Ilsung, Seoul, Korea). During surgery, lactated Ringer’s solution and 1 g of cephalosporin antibiotics were administered. Hair on the left facial side of the beagles was removed, and betadine and alcohol were used for sterilization. After exfoliating and exposing the mandible, the mandibular defect was induced according to the size of the model previously produced using the 3D printing technology.

Extraction was performed first, and it was implemented so that mandible is exposing. As the extracted part, the sagittal section of the mandible was prepared, and the inferior part was divided with an osteotome. It was separated into periosteo-elevator, and the pedicle vessels were preserved. Afterward, the pores at the bottom of the 3D-printed PCL/TCP/bdECm biomaterial were blocked with fibrin glue. Then, ADSC aggregates were injected into the pores of the 3D-printed model in the appropriate group, and fibrin glue was sprayed on the outer pores to prevent possible leakage of ADSCs. No treatment was provided for the PTE group.

The prepared 3D-printed model was inserted into the mandibular defect and fixed using a plate and screws. The skin was then sutured back (Figure 8). In both groups, the dressing was performed once every day from the day of surgery, and no dressing was done after the removal of the stitches on the 7th day. The study was approved by the Institutional Animal Care and Use Committee of Chonnam National University (Approval No. CNU IACUC-YB-2016-43, The approval date is 28 September 2016). It was conducted in compliance with the recommendations of the relevant committee.

### 4.3. Outcome Evaluation

#### 4.3.1. Evaluation Using CT

CT scans were performed on nine beagles immediately after surgery and before ossification progressed to evaluate the ossification of the 3D-printed model. Coronal, axial, and sagittal view CT images were obtained (Figure 7). At four and eight weeks after surgery, CT scans were taken again. Ossification activity of the implanted 3D-printed model was determined by evaluating the extent of calcification of the marginal area on CT, increase in bone density, and decrease in the size of the internal pores.

For the quantitative evaluation of ossification, bone density in five grafts of both groups was measured by Hounsfield unit (HU) after CT scan. The average HU was compared between the two groups [27].

#### 4.3.2. Histological Evaluation

All beagles were euthanized eight weeks after implantation of the 3D-printed model for histological evaluation. The grafts were excised, including 1 cm of the marginal bone tissue. A total of four histological evaluations were performed for each beagle. Two joints with normal bones and two tissues in the center of the scaffold were divided into quadrants. One quadrant was fixed in 10% neutral formalin and embedded in paraffin to make a tissue section. It was stained with Goldner’s trichrome stain. Under an optical microscope (SkyScan1173 (Ver. 1.6); Bruker-CT, Kartuizersweg 3B 2550 Kontich, Belgium), the extent of new ossification of the whole graft, thickness and level of periosteal formation, the extent of bone ingrowth into the pores, inflammatory cell infiltration of the graft, and formation of collagen fibers were observed. The optical density of each biopsy slide was measured using an image analysis system (i-SOLUTION LITETM, Image & Microscope Technology Inc., Cicero, NY, USA). In each group, the optical densities of the normal bone tissue, tissue in the pore, and soft tissue, including the periosteum formed around the scaffold, were randomly measured in five places to obtain an average [27].

#### 4.3.3. RT–PCR

Left-over tissues after histological evaluation were frozen with liquid nitrogen and crushed using a tea bowl. RNA was separated using Trizol and reverse-transcribed into cDNA using reverse transcriptase of a RevertAid first-strand cDNA synthesis kit (Thermo). Polymerase chain reaction (PCR) was subsequently performed using a PCR kit (Bioneer, Korea). Repetitive collagen type I (Col I), Osteocalcin, Runt-related transcription factor2 (Runx2) and glyceraldehyde-3-phosphate dehydrogenase (GAPDH) expression levels were measured in the RT–PCR amplified product (Table 1). The RT–PCR reactions were performed through 35 cycles of denaturation (94 °C, 45 s), annealing (62 °C, 60 s), and extension (72 °C, 60 s) for gene amplification.

#### 4.3.4. Western Blot

After histological evaluation, some of the remaining tissues were used to extract proteins for examining protein expression. The tissue was subject to reaction with PRO-PREP (iNtRON) lysis buffer for 30 min at 4 °C and centrifuged for 10 min at 13,000 rpm to observe the effects of bone differentiation induction at the protein level in each group. The extracted protein was quantified, heated at 100 °C for 5 min, and subject to electrophoresis in a 10% SDS-polyacrylamide gel. The extracted protein was transferred to a nitrocellulose membrane and isolated with 5% skim milk powder. The membrane was treated with primary antibodies, such as OCN (ab13420, 1 g/mL; Abcam, Cambridge, MA, USA), COL1 (ab6308, 1 g/mL; Abcam, Cambridge, MA, USA), RUNX2 (ab23981, 1 g/mL; Abcam, Cambridge, MA, USA), β-actin (ab8226, 1:10,000; Abcam, Cambridge, MA, USA). β-actin was used as a housekeeping protein. The membrane was then washed three times for 10 min with Tris-buffered saline Tween-20 (TBST) washing solution and reacted for two hours with horse-radish peroxidase-conjugated secondary antibody. The nitrocellulose membrane was washed three times again for 10 min using the same washing solution and reacted with ECL substrate solution for one minute. The membrane was then exposed to the X-ray film for development.

## 5. Conclusions

We implanted 3D-printed PCL/TCP/bdECM biomaterial into the artificial mandibular defect of beagles. ADSC aggregates were injected into the 3D-printed implants. The extent of ossification was analyzed via macroscopical assessment using 3D CT and histological and immunological stains. There were no immune rejections in the beagle model with scaffolds. Ossification was more abundant in the scaffold of those beagles that received ADSC aggregates.

In this study, macroscopical assessment using 3D CT and bone density measurement showed that ossification was relatively more pronounced in the beagles injected with ADSC aggregates. Additionally, bone density was also higher. The histological evaluation also demonstrated increased ossification in those that were administered with ADSC aggregates. RT–PCR and Western blot results also indicated that the levels of ossification proteins were relatively higher in the PTE + SA group.

This study confirmed that the fixation of bdECM-coated 3D PCL/TCP scaffold with ADSC aggregates could be an easy and effective technique for ossifying bone defect sites. A large-scale study is required to confirm the findings. We expect to develop 3D-printed tissue engineering and regenerative medicine through various attempts and collaboration with related systems.

## Figures and Tables

**Figure 1 ijms-22-05409-f001:**
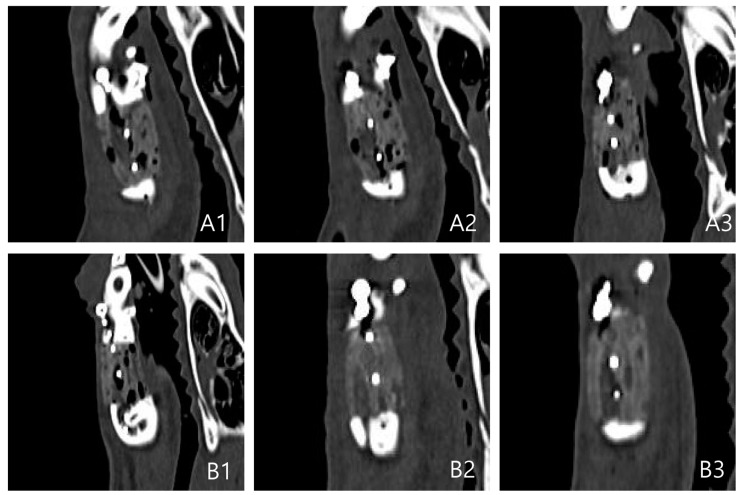
Sagittal views of CT. (**A**) PTE group. (**B**) PTE + SA group (1. immediate postoperative findings, 2. postoperative 4 weeks findings, 3. postoperative 8 weeks).

**Figure 2 ijms-22-05409-f002:**
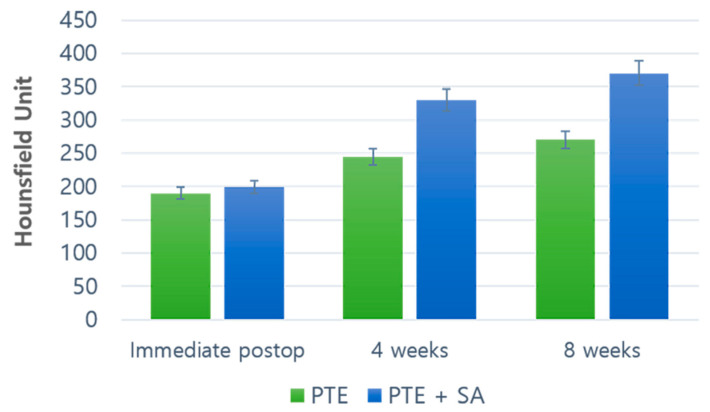
Hounsfield unit of the PTE group and the PTE + SA group.

**Figure 3 ijms-22-05409-f003:**
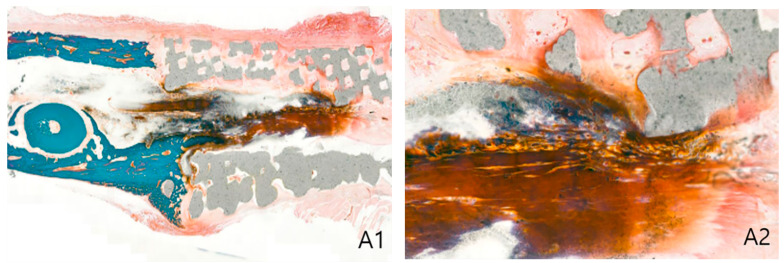
PTE group: biopsy of the 3D printing model of beagle 8 weeks postoperative day (Goldner trichrome stain). (**A**) Anterior marginal area, (**B**) posterior marginal area, (1. original magnification × 20, 2. original magnification ×100).

**Figure 4 ijms-22-05409-f004:**
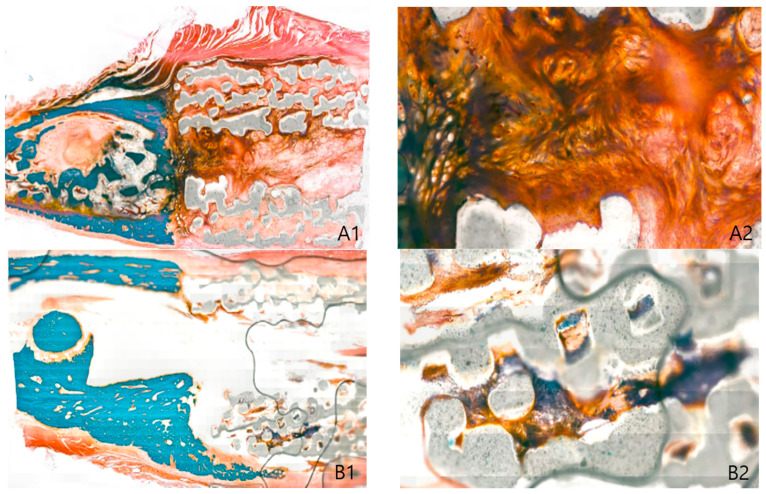
PTE + SA group: biopsy of adipose-derived stem cells with the 3D printing model of beagle 8 weeks postoperative day (Goldner trichrome stain). (**A**) Anterior marginal area, (**B**) posterior marginal area, (1. original magnification × 20, 2. original magnification ×100).

**Figure 5 ijms-22-05409-f005:**
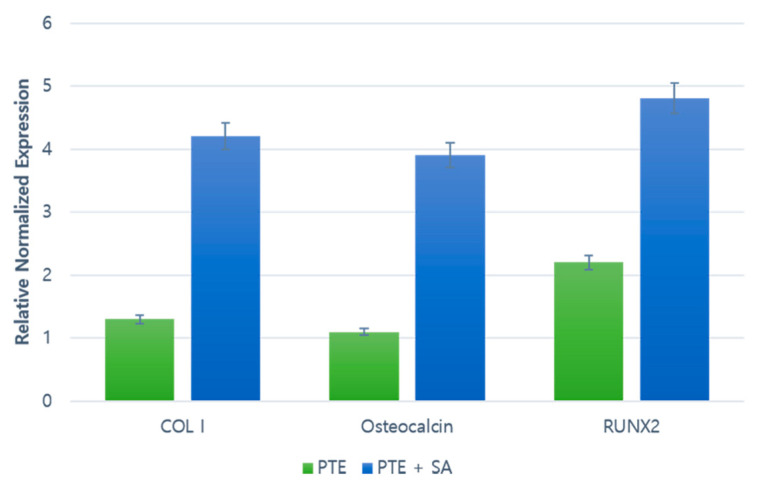
RT–PCR results of Collagen type I (ColI), Osteocalcin, and Runt-related transcription factor 2 (Runx2). A comparison of the gene expression of each protein at postoperative week 8.

**Figure 6 ijms-22-05409-f006:**
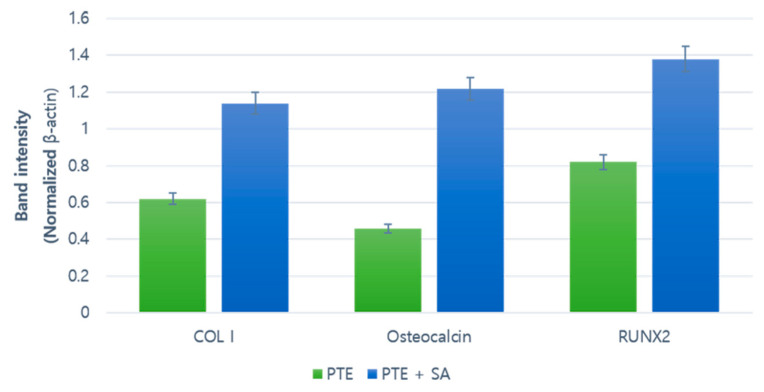
Western blot results of Runt-related transcription factor2 (Runx2), collagen type I (ColI), and Osteocalcin. A comparison of the protein analysis with electrophoresis at eight weeks postoperative.

**Figure 7 ijms-22-05409-f007:**
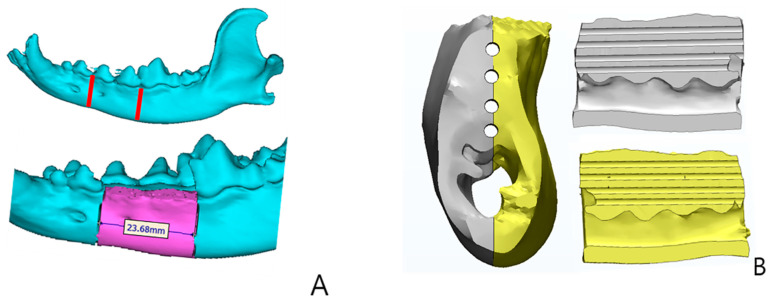
3D Plotter system. (**A**) Scaffold design, (**B**) PCL/TCP/bdECM 3D printing model (4 holes diameter 1 mm), (**C**) 4 point fixation using PCL + TCP pins.

**Figure 8 ijms-22-05409-f008:**
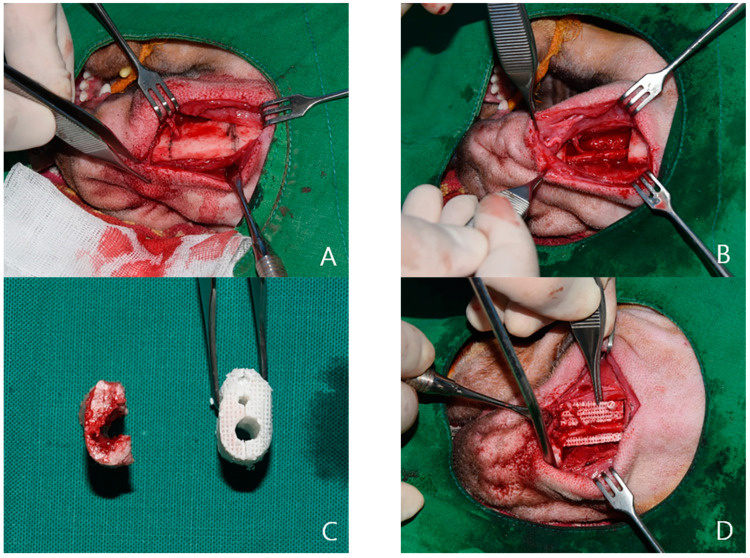
Experimental design. Design for mandibular bone defect of beagle model (**A**) after the defect was kept empty (**B**); PCL/TCP/bdECM 3D printing model (**C**); fibrin glue seeding after adipose-derived stem cells (ADSCs) was applied (**D**,**E**); fixed by plate and screw (**F**).

**Table 1 ijms-22-05409-t001:** Sequences of primers used for reverse transcription–polymerase chain reaction (RT–PCR). COL1, type 1 collagen; OCN, Osteocalcin; RUNX2, Runt-related transcription factor 2.

Gene Name	Sequence (5′-3′)
*COL1-dog-F*	CTCGTCACAGTTGGGGTTGA
*COL1-dog-R*	GGTGCAAGTATGAAGCGGGA
*OCN-dog-F*	AATTGCGCTCGAGCATCTCT
*OCN-dog-R*	ATTGCCACGGTTGCTACTGA
*RUNX2-dog-F*	GGCGGCTATAACTCTTCCCA
*RUNX2-dog-R*	ACGCAGCGGCTTTTTATTTCA
*GAPDH-F*	GTCGGAGTCAACGGATTTGG
*GAPDH-R*	GGGTGGAATCAATTGGAACAT

## Data Availability

The data presented in this study are available in the manuscript.

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
