# Peer review of "Osteogenesis of 3D-Printed PCL/TCP/bdECM Scaffold Using Adipose-Derived Stem Cells Aggregates; An Experimental Study in the Canine Mandible"

_ijms, 2021, doi:10.3390/ijms22115409_

Round 1

Reviewer 1 Report

The article entitled Osteogenesis of 3D-Printed PCL/TCP/bdECM scaffold using Adipose-Derived Stem Cells Aggregates; an experimental study in the canine mandible by Joon Seok Lee, Tae Hyun Park, Jeong Yeop Ryu, Dong Kyu Kim, Eun Jung Oh, Hyun Mi Kim, Jin-Hyung Shim, Won-Soo Yun, Jung Bo Huh, Sung Hwan Moon, Seong Soo Kang, Ho Yun Chung., describe an experimental study in the canine mandible where the 3D-Printed PCL/TCP/bdECM scaffold with Adipose-Derived Stem Cells Aggregates was used to sustain osteogenesis. Although the article is original, some experiments can be performed to improve this article, making it more interesting.

  1. Firstly, the authors should immunophenotype by FACS the isolated ADSCs for the main surface markers typical of the mesenchymal stem cells such as CD73+, CD90+, CD105+, CD34-.
  2. In the material and method section the authors should specify:
  • the growth medium for ADSCs;
  • the medium used to induce osteoblastic differentiation;
  • identify the antibodies used for WB;
  • the PCR program used specifying the temperature and the time for each cycle;
  • add primer sequences;
  • which housekeeping gene was used for PCR? please specify;
  1. Does 3D-Printed PCL/TCP/bdECM scaffold increase cell proliferation or stimulate osteoblastic differentiation with respect to plate culture? The authors could strengthen their article with molecular data that should have proceeded with the clinical application. They could compare cells plated on 3D-Printed PCL/TCP/bdECM scaffold with those cultured in plate in terms of:
  • cell proliferation and cell cycle;
  • expression of the main cells surface markers antigens;
  • differentiation potential towards osteoblastic lineage by evaluating ALP activity by NBT-BCIP assay after 5 days and red alizarin staining to show mineralized nodules after 20 days;
  1. The authors should complete the results section by showing western blotting with the expression of the Coll, Osteocalcin and Runx2 proteins suitably normalized with respect to a control protein.

The manuscript is well-written:  English language and style are fine, minor spell check is required

Reviewer 2 Report

Thank you for giving me the oppotunity to review the manuscript : Osteogenesis of 3D-Printed PCL/TCP/bdECM scaffold using Adipose-Derived Stem Cells Aggregates; an experimental  study in the canine mandible.

The manuscript presents a very interesting well conducted experimental study, however certan aspects need clarification.

  1. In the method section. The authors should describe in detail the bone defect induction surgery.
  2. Discussion section. The authors should compare the presented bone tissue engineering methods to other methods of bone engineering, using other scaffolds, other cell types.
  3. The authors should present the innovative aspect of their research, what the research brings as contribution to the present „know how”, knowing that there are other publication adressing this topic.

Liao HT, Lee MY, Tsai WW, Wang HC, Lu WC. Osteogenesis of adipose-derived stem cells on polycaprolactone-β-tricalcium phosphate scaffold fabricated via selective laser sintering and surface coating with collagen type I. J Tissue Eng Regen Med. 2016 Oct;10(10):E337-E353. doi: 10.1002/term.1811. Epub 2013 Aug 16. PMID: 23955935.

Rumiński S, Ostrowska B, Jaroszewicz J, Skirecki T, Włodarski K, Święszkowski W, Lewandowska-Szumieł M. Three-dimensional printed polycaprolactone-based scaffolds provide an advantageous environment for osteogenic differentiation of human adipose-derived stem cells. J Tissue Eng Regen Med. 2018 Jan;12(1):e473-e485. doi: 10.1002/term.2310. Epub 2017 Apr 11. PMID: 27599449.

Zanetti AS, Sabliov C, Gimble JM, Hayes DJ. Human adipose-derived stem cells and three-dimensional scaffold constructs: a review of the biomaterials and models currently used for bone regeneration. J Biomed Mater Res B Appl Biomater. 2013 Jan;101(1):187-99. doi: 10.1002/jbm.b.32817. Epub 2012 Sep 21. PMID: 22997152.

Round 2

Reviewer 1 Report

The article entitled Osteogenesis of 3D-Printed PCL/TCP/bdECM scaffold using Adipose-Derived Stem Cells Aggregates; an experimental study in the canine mandible by Joon Seok Lee, Tae Hyun Park, Jeong Yeop Ryu, Dong Kyu Kim, Eun Jung Oh, Hyun Mi Kim, Jin-Hyung Shim, Won-Soo Yun, Jung Bo Huh, Sung Hwan Moon, Seong Soo Kang, Ho Yun Chung., has been significantly improved so now it's to be publishbale in ijms.